# Machine Learning for Thyroid Cancer Detection, Presence of Metastasis, and Recurrence Predictions—A Scoping Review

**DOI:** 10.3390/cancers17081308

**Published:** 2025-04-12

**Authors:** Irina-Oana Lixandru-Petre, Alexandru Dima, Madalina Musat, Mihai Dascalu, Gratiela Gradisteanu Pircalabioru, Florina Silvia Iliescu, Ciprian Iliescu

**Affiliations:** 1eBio-Hub Centre of Excellence in Bioengineering, National University of Science and Technology POLITEHNICA Bucharest, 060042 Bucharest, Romania; irina.petre@upb.ro (I.-O.L.-P.); ggradisteanu@upb.ro (G.G.P.); florina.iliescu@upb.ro (F.S.I.); ciprian.iliescu@upb.ro (C.I.); 2Academy of Romanian Scientists, Ilfov 3, 050044 Bucharest, Romania; alexandru.dima1609@upb.ro; 3Faculty of Automatic Control and Computer Science, National University of Science and Technology POLITEHNICA Bucharest, 060042 Bucharest, Romania; 4Department of Endocrinology, C.I. Parhon National Institute of Endocrinology, 011863 Bucharest, Romania; 5Department of Endocrinology, Carol Davila University of Medicine and Pharmacy, 020021 Bucharest, Romania; 6Faculty of Biology, Department of Botany and Microbiology, University of Bucharest, 050095 Bucharest, Romania; 7Research Institute of University of Bucharest (ICUB), University of Bucharest, 050663 Bucharest, Romania; 8Faculty of Material Science and Engineering, National University of Science and Technology POLITEHNICA Bucharest, 060042 Bucharest, Romania; 9National Institute for Research and Development in Microtechnologies—IMT Bucharest, 077190 Voluntari, Romania

**Keywords:** thyroid cancer, machine learning, clinical data, prediction

## Abstract

In recent years, Machine Learning (ML) has achieved significant improvements in interpreting and evaluating medical data, especially regarding cancer. Without any apparent symptoms in its early stages, thyroid cancer exhibits a range of clinical, pathological, and molecular characteristics influenced by its histological subtypes. Current applications of ML models in thyroid cancer highlight their relevance in improving this disease’s diagnosis and management. This survey summarizes the current applications of ML in thyroid cancer, including methodologies and their performance. It also investigates how ML is used to develop simulation and prediction strategies based on clinical data and electronic medical records. This work serves as a reference for discussing the challenges and opportunities of integrating ML tools into routine endocrinology clinical workflows.

## 1. Introduction

Thyroid cancer is recognized as the most common malignant lesion of the endocrine glands [1,2]. In recent years, a steady rise in cancer incidence, coupled with the economic burden [3], has been observed worldwide, with the highest incidence in the fifth–sixth decades of life, and with a higher probability when more risk factors are present [4]. This situation is primarily due to the increased availability of more advanced and sensitive imaging technologies that facilitate the early diagnosis of small lesions, which is associated with the highest survival rates. However, the rising incidence of advanced thyroid cancers and the associated increase in mortality are also causes for concern. Ionizing radiation, obesity, and endocrine disruptors have the potential to trigger cancer onset in genetically susceptible individuals, including children. Advances made in identifying molecular subtypes of thyroid cancer allow for a personalized approach to the management of this disease.

According to the fifth edition of the World Health Organization’s classification of endocrine and neuroendocrine tumors [5], thyroid tumors are stratified into multiple categories. We will outline only the categories pertaining to malignant histological types relevant for the studies presented in this paper. The multiple categories are as follows:**Follicular cell-derived neoplasms**—Benign tumors, low-risk neoplasms, and malignant neoplasms. The malignant forms include the following:
Follicular thyroid carcinoma;Invasive encapsulated follicular variant papillary thyroid carcinoma;Papillary thyroid carcinoma;Oncocytic carcinoma of the thyroid;Follicular-derived carcinomas, high-grade:
–Poorly differentiated thyroid carcinoma;–Differentiated high-grade thyroid carcinoma;Anaplastic follicular cell-derived thyroid carcinoma.**Thyroid C-cell-derived carcinoma**—Medullary thyroid carcinoma.

Papillary (PTC) and follicular (FTC) carcinomas generally have a good prognosis, with 10-year survival rates exceeding 90%. Poorly differentiated thyroid carcinomas have a more severe prognosis, with 10-year survival rates of approximately 50%. In contrast, anaplastic thyroid carcinoma is frequently associated with 5-year survival rates below 10%, being the most aggressive form of thyroid cancer, as well as being rare and difficult to treat. The clinical protocol [1,4] for thyroid cancer generally consists of (a) presurgical diagnosis of thyroid nodules; (b) surgical removal of the primary tumor mass; (c) T (tumor) N (nodule) M (metastasis) system-based staging and other prognostic classifications; (d) Thyroid Stimulating Hormone (TSH) suppression therapy/hormonal therapy; (e) prevention/treatment of local/regional recurrences and distant metastases, i.e., post-surgery administration of radioactive iodine or, more rarely, chemotherapy, radiotherapy, immunotherapy, or a combination thereof; (f) follow-up through detecting tumor markers (e.g., thyroglobulin, antithyroglobulin antibodies), neck ultrasound, and whole-body scintigraphy.

Machine Learning (ML) is evolving rapidly, and the potential of Artificial Intelligence (AI)-based systems is increasingly recognized in the medical field. Therefore, this article surveys the existing related literature and assesses the role of ML models in thyroid cancer diagnosis and prognosis, focusing on predicting malignancy for improved early detection, or recurrence for improved treatment and quality of life. It highlights the ML methods used, discusses their performance, and compares them with traditional statistical and diagnostic procedures. Additionally, it emphasizes the advantages while addressing limitations, challenges, ethical considerations, and future directions toward personalized medicine.

Although advances in imaging, cytology, and molecular testing have improved thyroid cancer diagnosis and management, challenges such as misdiagnosis, overtreatment, unnecessary surgery, or insufficient patient-specific data persist [6]. ML tools can extract meaningful patterns and valuable information from multidimensional healthcare data, especially for cancer. Based on clinical data and EMRs for thyroid cancer, ML models excel in recognizing patterns, risk stratification, diagnostic support, or results prediction, offering opportunities to support evidence-based decision-making and improve thyroid cancer care [6]. Moreover, statistical analysis and modeling could examine the epidemiological indices of thyroid cancer via socioeconomic indices and reflect the TC distribution per country, learn the oncologic procedures’ outcomes, the patient experience, and QoL [7], and provide adequate economic epidemiology data to plan cancer prevention strategies [8].

Our main research question is to what extent current ML approaches support the shift toward personalized medicine in thyroid cancer. To answer this, we surveyed the literature on ML applications using patient data in key areas, including interpreting thyroid cancer diagnostics, predicting malignancy, improving thyroid cancer early detection, and predicting recurrence. We highlight the ML methods used, assess their performance through metrics like accuracy, Area Under the Receiver Operating Characteristic curve (AUROC), or F1-score, and compare them with traditional statistical and diagnostic methods. In addition, we discuss the advantages, limitations, challenges, ethical considerations, and future directions toward integrating ML into personalized thyroid cancer care. This contributes to the overall image created by systematic reviews focused on active surveillance, surgical intervention, remote access sites, open surgery, cost, complications, or recurrence-free survival rates. Therefore, addressing these aspects will contribute to the growing dialogue on integrating ML into clinical practice to improve thyroid cancer management.

## 2. Materials and Methods

This study reviewed all original research articles and conference papers at the intersection between medicine and informatics to analyze the current ML methods for thyroid cancer management using actual medical data and EMRs as a systematic data retrieval approach. The scoping analysis followed a previously established scoping review methodology, the PRISMA-ScR (Preferred Reporting Items for Systematic Reviews and Meta-Analyses Extension for Scoping Reviews Statement) guideline [9], and the study protocol was registered on INPLASY (INPLASY202520118). A comprehensive literature search was conducted to identify scientific studies that analyzed the connection between the term “thyroid gland” and information science. To identify relevant documents, on 7 November 2024, we utilized a query that searched for the title, abstract, and keywords of articles in six electronic databases (Scopus, Web of Science, Nature, Science Direct, Google Scholar, and PubMed) and reinforced multiple filters on the papers returned.

The following initial restrictions were selected during the paper selection: (1) the paper must be either a research article or a conference paper; (2) the paper publication date must be between 2014 and 2024; and (3) the paper must be written in English. The constructed queries searched for relevant word sequences (e.g., “machine learning”, “thyroid cancer”) in the titles, abstracts, and keywords to look for the papers that presented ML methods applied to thyroid cancer-related tasks (e.g., prediction, evaluation, or prognostic thyroid cancer disease). Scopus and Web of Science were relatively similar regarding the search queries utilized. For Scopus, we applied the following query: (TITLE-ABS-KEY(“deep learning”) OR TITLE-ABS-KEY(“machine learning”)) AND (TITLE-ABS-KEY(“thyroid cancer”) OR TITLE-ABS-KEY(“thyroid carcinoma”) OR TITLE-ABS-KEY(“thyroid sarcoma”) OR TITLE-ABS-KEY(“thyroid lymphoma”)), which yielded 486 distinct articles. Similarly, for Web of Science, the corresponding query was (TS = (“deep learning”) OR TS = (“machine learning”)) AND (TS = (“thyroid cancer”) OR TS = (“thyroid carcinoma”) OR TS = (“thyroid sarcoma”) OR TS = (“thyroid lymphoma”)), resulting in 55 articles, all of which were already found in Scopus. For ScienceDirect, the search interface did not provide a specific keyword mapping for title, abstract, and keywords. However, it allowed searches restricted to these fields. Consequently, the following query was employed within the “Title, Abstract, or Author-Specified Keywords” search textbox: ((“deep learning”) OR (“machine learning”)) AND ((“thyroid cancer”) OR (“thyroid carcinoma”) OR (“thyroid sarcoma”) OR (“thyroid lymphoma”)), which retrieved 73 articles. PubMed offered slightly less specificity as the query could be applied only to the title and abstract fields, yielding 326 articles. Nature and Google Scholar were the least specific among the sources analyzed as they did not allow for precise field-based searching. These databases returned 41 and 250 articles when using the same query. Afterward, the documents with fewer than 10 citations published between 2014 and 2022 and those with fewer than one in 2023 and 2024 were eliminated. Furthermore, the selected articles were filtered, and only those that used ML techniques applied exclusively the to actual medical data or electronic medical files of patients admitted to different hospitals or from known repositories (e.g., clinical and laboratory data) were kept. The studies were individually evaluated for eligibility, and titles were extracted in a narrative form and read full text.

The systematic search of multiple databases returned an initial screening of 1231 references, as illustrated in the PRISMA flow diagram from Figure 1.

In the initial screening phase, which involved reviewing abstracts or full texts, we retained only studies relevant to our research focus, specifically those utilizing real thyroid cancer medical data from hospitals or repositories. Articles were excluded for the following: (1) they only used patient imaging data; (2) they focused on applying machine learning to biomarkers or thyroid cancer-related genes. Following this screening, 203 papers were selected for further evaluation based on the eligibility criteria outlined in Table 1. Additional exclusions were made at this stage, including studies that predicted general thyroid disease rather than thyroid cancer specifically, as well as those examining the role of pollution in thyroid disease development. Ultimately, the 21 most relevant full-text articles were included and presented in this review, and the complete list is presented in Table 2. They were analyzed to identify and present the general purpose of the article with a focus on (1) the relationship between the use of ML in thyroid cancer; (2) the methodological aspects they selected; (3) the relevant information regarding their data (e.g., features, number of patients, the collection period); and (4) the ML methods used and their performance.

Additionally, we report the most representative metric for each reviewed study. To provide context, we briefly explain the primary evaluation metrics used in these studies: accuracy, AUROC, and F1-score. Accuracy measures the overall correctness of the model, AUROC evaluates a model’s ability to distinguish between positive and negative cases across thresholds, and the F1-score focuses on balancing precision and recall, being highly relevant to imbalanced datasets. We prefer reporting the AUROC or F1-score, if available, since they convey more information.

## 3. Results

### 3.1. Descriptives of Selected Studies

The data sources used in this study were clearly described. They were from public (Surveillance, Epidemiology, and End Results (SEER) database, Zenodo repository) or private repositories (hospitals/medical centers, universities). As to hospitals, we mention Shengjing Hospital, Peking Union Medical College Hospital (PUMCH), Xijing Hospital, Zhuzhou Hospital, Xiangya Medical College, Jinling Hospital, First People’s Hospital of Taicang, Chinese PLA General Hospital (China), Pusan (South Korea), King Fahad Specialist (Saudi Arabia), and as to universities, we mention Fourth Military Medical University, Nanjing University, Chongqing Medical University, Wenzhou Medical University (China), Friedrich Alexander University of Erlangen–Nuremberg (Germany), Yonsei University College of Medicine (South Korea).

Of the 21 studies, 17 were conducted in a single country, while the rest were collaborations between authors from two or more countries. Moreover, eight articles mentioned affiliations with China and six with the USA. Countries like Canada, Germany, and Korea were mentioned twice in the selected articles. In contrast, India, Iran, Saudi Arabia, or countries from Europe (the Netherlands, Poland, and Spain) were mentioned only once. It is worth noting that no paper was single-authored. Each paper had at least two authors, reflecting the diverse interactions and collaborative efforts between hospitals and universities, such as in the case of the USA, Canada, Poland, and Spain [13].

Furthermore, from 2014 to 2024, the number of articles published per year varied from one in 2014 to a maximum of seven in 2022. Figure 2 presents a time-based trend analysis chart with different statistical characteristics (e.g., countries involved, authors’ research collaborations, and most used ML methods). The trend is positive starting with 2019, and a considerable spike can be observed in 2021–2022 with the rise of machine learning/deep Learning techniques; however, as a limitation, additional studies may have been published in 2024 after the date of our web searches (November 2024).

Some studies included data about patients who underwent thyroidectomy [10,11,27], were diagnosed with thyroid cancer, or had different tumor features. Demographic characteristics and other clinical variables are available for all input data. The cohort studies spanned from a minimum of 1 year to a maximum of 19 years, and the size ranged from 50 to 51,291. The data shows that the most used ML techniques for the data type and study goals were as follows: (1) boosting models, applied 16 times in all 21 papers; (2) random forest, 14 times; (3) neural networks, 9 times; (4) logistic regression, support vector machine, and decision tree, 8 times each; (5) K-nearest neighbor, 5 times; and (6) Naïve Bayes, 3 times (Figure 2). The 21 publications with reference, thyroid cancer type, input data, paper objective, and best-performing classifier are presented in Table 2.

As such, the previous descriptive statistics, coupled with Figure 2, evidenced the most-used machine learning methods by diverse teams worldwide and the increasing interest in machine learning/deep learning techniques in medicine, highlighting the multidisciplinary aspect of AI-based medical research and the potential in thyroid cancer-focused healthcare.

### 3.2. Analysis of Selected Studies

This section presents a comprehensive overview of the ML-related methodologies and key findings, focusing on recent advancements in ML models for interpreting diagnostic data (e.g., EMRs and ultrasound imaging features). Upon researching the 21 selected articles, we identified three main approaches: (1) improvements in thyroid cancer detection through malignancy prediction and nodule classification [10,18,19,20,24,25]; (2) prediction of metastasis in the body derived from thyroid cancer [12,15,16,17,21,23,28,29,30]; (3) predicting recurrence and survival in thyroid cancer patients [11,13,14,22,26,27].

Each study below corresponds to a theme and includes a comprehensive overview of the methodologies and key findings, focusing on recent advancements in ML models for interpreting diagnostic data, such as EMRs and ultrasound imaging features. A notable advantage of ML in this domain lies in its capacity to uncover complex, nonlinear relationships between variables and patterns that may be challenging for human experts to detect. A detailed analysis and interpretation of these results is presented in Section 4. Please refer to Appendix A for a brief overview of the ML models and techniques introduced.

#### 3.2.1. Improving Thyroid Cancer Diagnosis Through Malignancy Prediction and Metastatic Nodule Classification

An important research direction involves utilizing preoperative or even pre-symp-tomatic markers to enhance the accuracy of predicting a patient’s likelihood of developing thyroid cancer. Additionally, these markers can aid in distinguishing between benign and malignant thyroid nodules, emphasizing early detection and comprehensive risk assessment.

Olatunji et al. [19] employed ML techniques for the early detection of thyroid cancer at pre-symptomatic stages. The authors used data from 218 patients at King Fahad Specialist Hospital, with an equal distribution of those diagnosed with thyroid cancer and other diseases. The resulting dataset consisted of 14 attributes, primarily derived from blood test parameters (e.g., hematocrit, alanine transaminase, and red blood cell count) and demographic variables, namely, age and sex. They experimented with various classical ML algorithms, including random forest (RF) [31], support vector machine (SVM) [32], naive Bayes [33], and multi-layer perceptron (MLP) [34]. Among these models, the RF classifier obtained the highest performance, achieving an F1 score of 96%. However, most data originated from papers on clinical aspects such as pre-, postoperative, invasive, and non-invasive investigations. Since cancer diagnosis is complex and involves non-invasive and invasive modalities, it is essential to gather data and design adequate schemes to increase patient compliance and the rate of successful detection and comprehensive follow-up.

Xi et al. [10] improved the accuracy of thyroid nodule malignancy prediction with the help of a constructed and publicly released patient records dataset from Shengjing Hospital of China Medical University. The study covered thyroidectomy cases from 2010 to 2012. Unlike previous studies, the dataset included standard demographic and biochemical markers (i.e., FT3, FT4, TSH, TPO, TGAb) and detailed ultrasound-derived characteristics of thyroid nodules (e.g., location, multifocality, size). Therefore, it provided a more comprehensive feature set for analysis. With this dataset, comprising records from 724 patients and 1232 nodules, the researchers evaluated multiple ML models, including gradient boosting [35], Logistic Regression (LR) [36], SVM, and RF. The RF model demonstrated the highest predictive performance, achieving an AUROC of 85.41 and significantly surpassing expert assessments in malignancy detection. The study also identified calcification, laterality, blood flow, and nodule location as the most influential predictors of malignancy.

Furthermore, Sankar and Sathyalakshmi [18] proposed integrating association-rule mining algorithms with classical ML algorithms to distinguish between malignant and benign thyroid nodules. Their study used the previously introduced dataset [10] and initially applied data cleaning, normalization, and standardization (i.e., continuous numerical values were converted into categorical variables). Subsequently, they employed the Apriori algorithm and FP-growth [37,38] to extract association rules for malignant and benign classes, ranking the rules based on multiple evaluation metrics. Following the rule extraction process, they trained and evaluated various classical ML algorithms on the preprocessed dataset, including RF, XGBoost [39], SVM, and Decision Tree (DT) [40]. To further enhance feature selection and interpretation, the authors applied SHapley Additive exPlanations (SHAP) [41] to assess the importance of individual features. Combined with the ranked association rules, they identified and selected high-ranking or dominant features derived from both methods. The identified set of rules was then integrated back into the dataset. Finally, the ML models were trained on the refined dataset to incorporate the most relevant features and optimize classification performance. They achieved a high F1 score of 95% and identified the most influential features impacting model performance. Calcification and shape emerged as the most significant indicators of thyroid malignancy. Size, site, and multifocality contributed positively to malignancy prediction, while multilateralism, margin, and blood flow negatively impacted malignancy classification.

Since cancer diagnostic schemes comprise invasive and noninvasive modalities, their complexity is correlated with interpatient and intrapatient variability. Controlling the outcomes could come from combining medical and AI-based technology. In this direction, Luong et al. [25] investigated the application of ML for predicting malignancy in indeterminate thyroid nodules using non-invasive test data. Approximately 30% of the thyroid nodules examinations via fine-needle aspiration biopsy (FNAB) resulted in indeterminate results, leading to surgical pathology for definitive diagnosis. Therefore, the study aimed to develop a predictive model to reduce the need for unnecessary surgeries by leveraging data from less invasive diagnostic methods. A retrospective analysis was conducted on 489 indeterminate thyroid nodules with cytology reports, each accompanied by at least one corresponding pathology report from the same patient. After inclusion criteria were applied, requiring an indeterminate cytologic result and a completed diagnostic pathology report, 355 nodules were included in the final dataset, of which 48.2% were confirmed malignant. The EMRs-based dataset integrated demographic, cytopathological (e.g., age at first FNAB, calcification, FNAB result), and ultrasound data. Eight ML classifiers were tested using repeated stratified k-fold cross-validation (10 folds, 10 repeats) with default hyperparameters to evaluate malignancy prediction performance. The models assessed included ridge regression [42], naive Bayes, SVM, RF, and gradient boosting. Among these, the highest performance was obtained with RF, with an AUROC of 85.9%.

On the other hand, Sievert et al. [20] distinguished their approach from previous methodologies by relying exclusively on textual reports of preoperative ultrasound examinations to predict malignancy risk based on the Thyroid Imaging Reporting and Data System (TI-RADS). The dataset consisted of 50 anonymized clinical reports and corresponding risk assessments validated through histological outcomes of thyroidectomy procedures. Furthermore, in contrast to prior studies, they employed a Large Language Model (LLM), specifically ChatGPT-3.5, to classify thyroid nodules, marking a novel application of LLMs in this domain. Overall, the model had an accuracy of 42% in identifying the correct TI-RADS category. Unfortunately, the study has several notable limitations, such as a small sample size, the absence of a well-defined answering methodology from the LLM, which introduced excessive variability, and an outdated ChatGPT version. These factors collectively diminished the study’s reproducibility and reliability.

Unlike the previous studies, Radebe et al. [24] sought to develop an ML model for predicting malignancy in pediatric patients diagnosed with PTC. The study reviewed 198 patients with thyroid masses, excluded cases of prior thyroidectomy and missing data, and proposed for analysis a final cohort of 140 patients with complete records, of whom 69 were confirmed malignant nodules. An RF model with inTrees [43] was employed to generate interpretable rule sets. The model was trained on a private dataset and incorporated clinical and sonographic features such as demographic information, ultrasound findings, and biopsy results. With an AUROC of 83.87%, the rule-based model significantly outperformed historical diagnostic approaches in identifying nonbenign cytology and malignant histology, which had an AUROC of 58.93%. This enhancement aids clinicians in determining which patients who have not yet undergone thyroidectomy may benefit from biopsy or surgical intervention.

As an overview, this section explored the use of ML to enhance early thyroid cancer detection, improve malignancy prediction, and reduce unnecessary surgical procedures. Various ML models, including random forest, support vector machine, and gradient boosting, have been applied to diverse datasets containing clinical, biochemical, and ultrasound-derived features. Random forest consistently demonstrated high predictive performance across multiple studies. Additionally, advanced methods like association rule mining and SHAP explanations have been integrated to improve interpretability. While ML-based approaches have shown promise in refining diagnostic accuracy, challenges remain, such as dataset limitations, variability in feature selection, and the need for further validation in real-world clinical settings.

#### 3.2.2. Identifying Secondary Metastases Arising from Thyroid Cancer

Another significant research direction we have identified involves detecting and characterizing metastatic thyroid cancer. Accurately identifying metastases is essential for treatment planning and prognosis assessment, as metastatic thyroid cancer often requires more intensive therapeutic approaches. Therefore, studies highlighted means of modeling for metastasis analysis. For instance, Liu et al. [12] investigated the efficiency of using classical ML models in predicting lung metastasis in patients with thyroid cancer. The study used the open-source SEER database [44], which included a wide range of patient attributes, such as demographic variables (e.g., age, sex, race, marital status) and morphopathological characteristics (e.g., location, grade, number of lymph nodes, tumor size, number of positive nodes, presence of metastases, histologic type). The authors focused on patients diagnosed between 2010 and 2015 and selected only the following features: age, sex, race, grade, TNM stage, race, laterality, and survival months. They experimented with several models, such as SVM, XGBoost, and K-nearest neighbours [45]. The random forest model achieved the highest performance, with an F1 score of 72% outperforming the other approaches. The findings indicated that lung metastasis risk varies with age; it is lower in patients aged 20 to 40 and increased in patients after 60. Higher T and N stages and increased tumor grade were also associated with an elevated risk of lung metastasis. Among thyroid cancer subtypes, PTC exhibited the lowest risk, whereas anaplastic thyroid carcinoma (ATC) had the highest. A notable limitation of the study was the class imbalance in the dataset, as thyroid cancer metastasis cases were relatively sparse. The authors employed synthetic oversampling techniques to address this issue, improving model training and enhancing predictive performance.

Moreover, detecting metastasis faces several other issues. For instance, bone scintigraphy, traditionally used to detect bone metastasis, is costly, carries radiation risks, and increases the risk of tumor cell proliferation. To better address these problems and provide a non-invasive alternative, Liu et al. [15] developed a machine-learning model to predict the risk of bone metastasis in patients newly diagnosed with thyroid cancer. The study used data from the SEER database (2010–2016), including 17,138 thyroid cancer patients, among whom 166 (0.97%) developed bone metastasis. The study included features like those used by Liu et al. [12], including additional demographic variables. They applied a random forest classifier, excluded samples with incomplete data, and achieved an AUROC score of 91.7%. The study’s limitations stemmed from the restriction to a North American population, which may have limited the model’s generalisability. Additionally, SEER’s lack of post-diagnosis treatment data meant that treatment effects on bone metastasis risk could not be analyzed. These limitations highlight the need for broader datasets in future research to support clinical needs.

In line with the clinical requirements is the study by Lai et al. [21]. It developed a web-based ML tool to enhance the predictive accuracy of lateral lymph node metastasis (LLNM) in patients with PTC, thereby supporting clinical decision-making. The study used retrospective clinical, laboratory, and ultrasound data from 1,815 PTC patients who underwent primary thyroidectomy at the Chinese PLA General Hospital between 2015 and 2020. LLNM was observed in 62.53% of cases. To develop the predictive model, the authors experimented with various classical ML algorithms, such as DT, RF, XGBoost, SVM, Neural Networks (NN), and K-nearest neighbours, achieving an AUROC of 80%. The study also implemented multiple data preprocessing techniques, such as handling missing data, detecting outliers, and feature selection using LASSO [46]. Given that most patients were diagnosed with LLNM, the authors applied the Synthetic Minority Over-sampling Technique (SMOTE) [47] to balance the dataset. To enhance model interpretability, they employed SHAP values, identifying the most influential variables in predicting LLNM, including tumor size, lymph node microcalcification, patient age, lymph node size, and tumor location. Despite its predictive potential, the model has certain limitations, particularly the need for more extensive prospective studies to refine its applicability. Specifically, the study used a cutoff age of 55 years, aligning with the 8th edition of the American Joint Committee on Cancer (AJCC) staging system. However, further research is needed to clarify the impact of age on tumor progression and ensure broader clinical relevance.

Epidemiological studies could supply enough data and support with demographics and clinical information that could aid in assessing risk factors and predictive values in the studied populations. Moreover, they could positively impact the medical outcomes and the quality of life if coupled with sensitive tools and carefully designed diagnostic and therapeutic schemes. Another retrospective study was conducted by Wu et al. [17] on patients with PTC. This study intended to create ML models to accurately predict central lymph node metastasis (CLNM). It is already acknowledged that definitive CLNM diagnosis primarily relies on postoperative pathology, while the preoperative risk factors remain poorly understood. The study included 1,103 patients who underwent initial thyroid resection between 2018 and 2019, with a final pathological diagnosis of PTC. Patients with other thyroid tumor types, those who had received chemotherapy or radiotherapy for thyroid malignancy before surgery, and those with incomplete clinical data were excluded. The dataset consisted of 22 variables: demographics, clinical (e.g., BMI, T3, T4, TSH), and ultrasonography characteristics. Several ML algorithms were tested, including RF, MLP, Gradient Boosting Decision Tree (GBDT) [35], and AdaBoost [48]. The results indicated that GBDT, MLP, and RF consistently outperformed other models in CLNM prediction, with GBDT achieving the highest performance, attaining an AUROC of 73.1%. Key predictive features included suspected lymph nodes, tumor size, age, micro-calcifications, gender, TPO-Ab, TSH levels, tumor shape, hypoechogenicity, and capsular invasion. The findings suggested that CLNM is more likely to occur in younger patients, emphasizing the need for a larger cohort of diagnosed PTC cases to enhance model validation. Another study drawback is the limited reproducibility due to the high prevalence of microcarcinomas (≤1 cm), which are not routinely surgically treated in many centers.

Another study on models for CLNM in PTC was conducted by Yu et al. [23]. The work was on predictive models for CLNM in clinically node-negative papillary thyroid microcarcinoma to facilitate real-time personalized surgical decision-making. The team constructed a dataset with data gathered from a cohort of 1,121 patients, of whom 33.5% were diagnosed with CLNM. The dataset included clinical, pathological, and ultrasound-derived features. ML models, such as multivariate adaptive regression splines (MARS) [49], RF, XGBoost, and MLP, were evaluated. The AUROC values on the validation set were between 66.4% and 79.4%, with the RF attaining the highest predictive performance. Since the study relied on biopsy and postoperative data for certain variables, the model’s applicability in preoperative settings was restricted. Additionally, the absence of long-term follow-up data limited the study’s ability to verify individual prognosis. Based on the general acknowledgment that a performant model could facilitate the early identification of high-risk patients, more data are required to design strategies and eventually enable personalized therapeutic approaches such as targeted surgery or disease surveillance. For instance, assessing CLNM risk is crucial for determining the extent of thyroidectomy and lymph node dissection and is helpful when minimizing overtreatment and associated complications are observed. Therefore, less invasive treatment options may be considered for low-risk patients.

It is worth mentioning the work of Zhu et al. [16] on ML models to predict CLNM in clinically low-risk PTC patients, specifically those with T1-T2 stage, non-invasive, clinically node-negative tumors based on ultrasound imaging. This subset of patients has been largely understudied, making their research particularly relevant. Toward this goal, they analyzed data from 1271 patients who underwent thyroid surgery between 2016 and 2018, excluding those with distant metastasis, prior thyroid surgery, or incomplete clinical information. The dataset comprised demographic features, tumor information (e.g., size, bilaterality, location), and associated conditions (e.g., Delphian lymph node involvement). The study experimented with classical ML algorithms, such as LR, XGBoost, RF, DT, and a small MLP. XGBoost demonstrated the highest predictive performance, achieving an AUROC of 75%.

More data to support this line of research came from Wang et al. [28]. The group focused on developing an effective predictive model for CLNM in patients with PTC and collected data from a cohort of 488 PTC patients diagnosed via ultrasound-guided fine-needle aspiration biopsy. The dataset included demographics, pathomorphological (e.g., sex, age, BRAF mutation status, lymph node metastasis), and ultrasound characteristics (e.g., nodule location, microcalcification, diameter). Independent risk factors for CLNM were identified through binary logistic regression analysis along its associated nomogram model, including age (younger patients had a higher risk), maximum diameter of thyroid nodules (larger nodules increased the risk), capsular invasion (presence was a strong predictor), and BRAF V600E gene mutation (highly significant in predicting CLNM). The model achieved an AUROC of 76%. Additionally, the trained six-layer one-dimensional Convolutional Neural Network (CNN) achieved an AUROC of 78%, demonstrating a stronger predictive performance.

Regarding metastasis and related surveillance in PTC, Zhu et al. [29] investigated the feasibility of developing an effective predictive model for skip metastasis for PTC patients. Skip metastasis refers to lateral lymph node metastasis without the central neck compartment involvement. The authors highlighted that PTC with lymph node metastasis can lead to local recurrence and cancer-specific mortality in some patients. Moreover, approximately 30% of metastatic lymph nodes remain undetectable in preoperative examinations, underscoring the need for a reliable predictive model. Therefore, the study analyzed clinical data from 18,192 thyroid cancer patients between 2016 and 2020 and identified 820 cases with confirmed skip metastasis by the postoperative pathological report. The dataset included demographics, clinical variables (e.g., BMI and associated diseases), and ultrasound characteristics (e.g., tumor size, location, margins). The SVM-based analysis achieved an AUROC of 72.1%. The skip metastasis rate was 13.3% in the training and 11.1% in the validation samples. However, the study’s interpretability was limited by the chosen performance metrics. Given the significant class imbalance, alternative measures such as the F1-score would have provided a more comprehensive model performance evaluation.

Moreover, Zhou et al. [30] aimed to develop a preoperative ultrasonography-based predictive model for Delphian lymph node metastasis (DLNM) in patients with PTC. The study used patient data collected from January 2014 to December 2021 at Jinling Hospital to construct a training dataset comprising 316 patients with 402 thyroid lesions. A validation dataset was also obtained from the First People’s Hospital of Taicang, consisting of 280 patients with 341 lesions. The dataset included a comprehensive set of variables, incorporating demographic and clinical factors (e.g., age, sex, blood test results), ultrasound-derived features (e.g., nodule size, location, morphology), and pathological characteristics (e.g., multifocality, specimen size). The researchers evaluated two ML models, RF and multivariate logistic regression, and identified that RF yielded the highest predictive accuracy of 87.39%. Their findings indicated that CLNM and LLNM are associated with DLNM, aligning with existing literature. Furthermore, the study identified serum calcitonin as a novel factor linked to DLNM, providing further insights into potential predictive markers for thyroid cancer metastasis.

Overall, this section explored the use of machine learning for detecting and characterizing metastatic thyroid cancer, which is crucial for prognosis and treatment planning. Researchers have applied models to predict metastasis in various thyroid cancer subtypes. Random forest and XGBoost consistently achieved strong predictive performances, with studies identifying key risk factors like tumor size, lymph node involvement, age, capsular invasion, and genetic mutations (e.g., BRAF V600E). Techniques such as SHAP values and oversampling improved model interpretability and performance. However, challenges persist, including dataset imbalances, generalizability issues, and reliance on postoperative data, limiting real-time clinical application.

#### 3.2.3. Predicting Recurrence and Survival in Thyroid Cancer Patients

It is already acknowledged that the timely detection of malignancies is crucial in early and personalized treatment. This section presents the potential of ML as a prognostic tool in well-differentiated thyroid cancers, particularly for the papillary and follicular types. Several papers focused on cancer prognosis using ML algorithms. For instance, Mao et al. [27] worked on predicting FTC progress. The study utilized data from the SEER database and analyzed 6,891 patients diagnosed with primary FTC between 2004 and 2015. ML models such as XGBoost, LightGBM [50], LR, SVM, naive Bayes, and MLP were evaluated, with XGBoost achieving the highest predictive performance with an AUROC of 90.4%. The study identified age, marital status, T classification, N classification, M classification, and surgical methods as independent risk factors for cancer-specific survival. The Kaplan–Meier method [51] and Cox regression model [52] were additionally employed to analyze survival risk factors further, and SHAP values were incorporated to enhance model transparency and interoperability. A key finding revealed that marital status significantly influenced survival outcomes, with married patients having a survival advantage over widowed or divorced individuals. This observation underscored the importance of considering socio-demographics in cancer prognosis and patient care. Moreover, the choice of treatment was found to be a critical prognostic factor. While total thyroidectomy has traditionally been regarded as the standard treatment for FTC, concerns over its associated complications have led to reconsideration. The results showed that unilateral thyroid lobectomy with isthmectomy provided comparable, or in some cases, superior prognostic outcomes for selected patients.

Another observed and investigated aspect was related to clinical and pathomorphological factors associated with PTC. For instance, Park and Lee [11] developed an ML-based prediction model for cancer recurrence. The study retrospectively analyzed medical records of 1,040 patients diagnosed with PTC between 2003 and 2009 who underwent either total or partial thyroidectomy. Patient data, including demographic, clinicopathological, and ultrasound-related parameters, were used to construct a dataset for model training. Patients with distant metastases at diagnosis, a history of prior head or neck surgeries, or incomplete follow-up data were excluded. Among the total cohort, only 41 patients experienced cancer recurrence. Therefore, SMOTE was applied to address the class imbalance. The authors used DT, RF, XGBoost, LightGBM, and an ensemble stacking approach. From the included factors, only sex and tumor size were significantly correlated with disease recurrence. Additionally, lymph node ratio, CLNM characteristics, and tumor size were the most influential in the best-performing classifier. Their best model, DT, only obtained a maximum F1-score of 28%, highlighting the challenge of accurately predicting recurrence in this dataset.

Another dataset compiled by Kim et al. [22] comprised clinical, pathological, genetic, and laboratory data and explored the practical application of inductive logic programming (ILP) for predicting disease recurrence and prognosis in patients with well-differentiated thyroid cancer with thyroidectomy. The study analyzed a cohort of 785 thyroid cancer patients who underwent bilateral (total) thyroidectomy, received radioiodine treatment, and were followed for more than five years. The study employed ILP through the DELMIA Process Rules Discovery tool [53] to develop predictive algorithms. Eight rules were extracted: five for predicting non-recurrence and three for predicting recurrence. The identified rules correctly predicted 71.4% of recurrence cases within the model group. However, it is essential to note that the rules did not encompass all patients in the dataset, leaving some cases undetected for both groups. The most influential parameters in the model included postoperative thyroglobulin levels, BMI, anti-thyroglobulin antibody levels, TSH levels, tumor size, and lymph node metastasis. While ILP was characterized by its strong predictive performance and the advantage of interpretable rule generation, the limited sample size of recurrence cases limited the generalizability and validity of the proposed approach.

Yang et al. [26] developed a novel ML-based approach, the ensemble algorithm for clustering cancer data, to enhance prognostic systems for PTC and FTC. The study aimed to facilitate the transition from traditional cancer staging systems, such as the AJCC Cancer Staging Manual (8th edition) [54], to ML-driven models, ensuring greater adaptability among healthcare providers. The researchers used patient data from the open-source SEER database (2004–2010) and investigated three feature sets. The first feature set included primary tumor type, regional nodes positive, metastasis, and age. The second set added histological cancer type to the first set of variables, while the third was similar to the first but employed multiple age cutoffs. The study analyzed patient survival over the next five years and implemented a clustering approach. It applied Partitioning Around Medoids (PAM) [55], followed by hierarchical cluster analysis. The model demonstrated superior predictive accuracy compared to the AJCC 8th edition staging system. Specifically, it achieved an AUROC of 85.83%, outperforming the AJCC model, with an AUROC of 83.87% in stratification and survival prediction based on tumor size (T), regional lymph nodes (N), distant metastasis (M), and age (A). The analysis identified 55 years as the optimal age cutoff, while the feature set incorporating histologic type yielded the highest prediction accuracy. While the model’s groupings correlated strongly with AJCC staging, its superior predictive performance suggested that it can provide enhanced prognosis ability and aid clinical decision-making for thyroid cancer patients.

Another study employed the SEER database to explore ways to address the long-term survival of patients with PTC and FTC. Mourad et al. [13] used the SEER database (1988–2007) and aimed to enhance the prediction capability regarding long-term survival (i.e., 10+ years) of PTC and FTC patients. The study incorporated a range of clinicopathological and ultrasound-derived variables to train three distinct MLP models. The primary distinction between the MLP models is the feature selection strategies used during training. The first model used seven variables: age, race, gender, tumor size, primary disease extent, location of nodal disease, and the number of positive lymph nodes. The second model was trained on a reduced set of three variables, selected through feature selection algorithms, specifically Fisher’s discriminant ratio [56], Kruskal–Wallis analysis [57], and relief-F [58]. The third model incorporated only TNM staging features. The highest predictive performance was achieved using the first model with an F1-score of 43.1%. This aspect highlights the difficulty of accurately predicting the long-term survivability of PTC patients. Despite problems, evaluating the MLP model further and comparing it with other approaches proved useful in short-term survival rate prediction.

Jajroudi et al. [14] aimed to enhance ML models for survival prediction at 1-year, 3-year, and 5-year intervals in patients with thyroid cancer. Their research used a dataset of 7706 samples with 16 attributes extracted from the SEER database. The study evaluated the performance of both an LR and an MLP model. The findings indicated that MLP outperformed LR in 1-year survival prediction, achieving an accuracy of 92.9% compared to 81.2%. However, for longer-term survival predictions, LR had superior performance, with accuracy rates of 88.6% versus 85.1% for the 3-year prediction and 90.7% versus 86.8% for the 5-year prediction. However, the study acknowledged certain limitations, primarily due to missing data for potentially influential features such as “lymphoma subtype record” and “lymphovascular invasion”, which had to be excluded due to insufficient data availability. This omission may have affected the overall predictive performance of the models. Many studies demonstrated interest in AI-based tools with clinical potential for personalized medicine. To date, there is a need for more data from continuous research to consolidate the initial conclusions, select the best solutions, and translate them into various clinical applications.

This section has explored the potential of using machine learning as a prognostic tool for well-differentiated thyroid cancers, particularly papillary and follicular types. Various models have been used to predict cancer progression, recurrence, and long-term survival rates. Studies utilizing the SEER database and hospital records have identified key predictors. While ML models outperform traditional staging methods like AJCC 8th edition, challenges such as class imbalance, limited recurrence cases, and missing data impact predictive accuracy.

A summary containing the highlights of the previous sections is presented in Box 1.

Box 1Highlights from the selected studies.

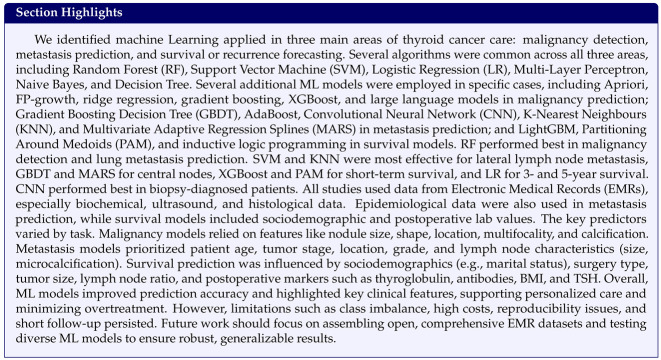



## 4. Discussion

### 4.1. Evidence Synthesis

Despite the existence of established protocols [1,4], thyroid cancer continues to pose significant challenges regarding accurate diagnosis, risk stratification, and personalized treatment. Integrating ML into thyroid cancer research and management holds transformative potential, particularly when applied to clinical data and EMRs. By leveraging these data sources, ML algorithms can enhance diagnostic precision, predict disease progression, survival, and recurrence, and optimize therapeutic decision-making. Furthermore, ML could improve medical practice through AI-driven decision support systems and clinical applications for risk stratification and treatment planning. It can improve diagnostic accuracy by identifying subtle risk factors and reducing clinician variability in interpreting diagnostic imaging and histopathology, thus minimizing overtreatment and its associated complications. Incorporating relevant variables, such as genetic data [59], into clinical practice and developing user-friendly AI tools that integrate seamlessly into electronic health records for real-time decision-making would significantly enhance the management of thyroid cancer.

However, several challenges remain, requiring consensus and further research. It is essential to assess the convergence of the current state of thyroid cancer treatment and the role of intelligence-based methods in addressing the related complexities. For instance, an accepted and commonly used workflow paradigm includes steps like:(a)Dataset acquisition—sourcing and collecting relevant data;(b)Data preprocessing and dimensionality reduction—cleaning data, exclusion sets, normalization, standardization;(c)ML model training and disease prediction—also implying feature selection, grouping methods, applied ML techniques;(d)Model evaluation—assessing model performance using predefined and well-documented metrics.

It is worth mentioning that all 21 selected articles followed the same working method. Regarding the dataset acquisition, the data sources were clearly described, coming from repositories, hospitals/medical centers, or universities. However, in the second step, the model will be affected if data preprocessing differs due to the missing values. Some authors solved this issue by eliminating attributes with missing values greater than 50%. In contrast, others excluded all patients with missing or insufficient data, with unknown clinicopathologic profiles, or undetermined histology [14,15,24]. In most works, missing data have been excluded, and noisy data were handled by choosing models suitable for their resilience against noise [15]. Step-like feature selection was presented and mentioned in articles with algorithms like SHAP (Shapley Additive Explanations), Fisher’s discriminant ratio, Kruskal–Wallis’ analysis, or relief-F [10,13,17,18,21].

Upon comparing the papers, it was observed that the most employed ML techniques for the given data types and study objectives were boosting models, random forest, and artificial neural networks. Furthermore, each paper calculated performance metrics like the accuracy, AUROC, or F1-score of the chosen ML algorithm(s) in the validation set, considering statistically significant (*p*-value <0.05) and the relative importance of the most important predictors of the model(s) is summarized. Also, 8 of the 21 reviewed studies incorporated ultrasound features in addition to standard clinical data [10,16,17,21,24,25,29,30], while other studies compared the performance of ML models with conventional methods [21,26,27], where the authors evaluated their findings against the traditional AJCC staging system, achieving comparable or superior results.

### 4.2. Principal Challenges and Limitations

While this study comprehensively overviewed the perspectives of integrating ML in thyroid cancer research, several limitations must be acknowledged. The observed constraints primarily relate to the selection of data sources, publication types, language restrictions, and the defined time frame, which may have influenced the scope and completeness of the findings. We outline the key limitations of our study and their potential impact on the results as follows. One of the primary limitations is related to data retrieval and information source selection. This literature search used six online databases to identify relevant papers published between 2014 and 2022 on the targeted topic (i.e., thyroid cancer and Machine Learning). While the databases provided a substantial research collection, other repositories may contain additional relevant studies. Another significant limitation pertains to the type of publications included. Our methodology specifically filtered for research articles and conference papers with full-text availability. Consequently, other forms of academic contributions, such as book chapters, brief reports, letters, editorials, and case studies, were not considered. Access restrictions posed another constraint, as publications requiring a subscription or purchase were excluded, limiting our ability to assess their content fully. Language restrictions also represented a limitation, as the search was restricted to English-published articles. Studies published in other languages were intentionally omitted, potentially excluding valuable research contributions from non-English sources. Lastly, the time interval of the study may be considered a constraint, as only publications from 2014 onward were included. However, even though an extended time frame might seem beneficial, it may have been minimal due to the relatively small number of articles published before 2014 and the employed methodologies already integrated by the recent works included in our analysis.

So far, we have discussed our study’s limitations. We now turn to the limitations identified in the reviewed articles. Since limitations often arise from broader challenges in the field, we highlight several recurring issues and obstacles reported across the literature that may hinder the development and implementation of effective ML applications in thyroid cancer care.

For instance, Sankar and Sathyalakshmi [18] highlighted the principal challenges and concepts related to model evaluation and bias-variance trade-offs in the context of TC classification using ML. The error model sources stemmed from the bias level, the model variance, or the variance of the irreducible error in the data. Bias indicates how well the model fits the training dataset. For example, a high bias level can result in underfitting, meaning that the model oversimplifies the data and their underlying complexity, resulting in sub-par performance. The model’s variance describes the model’s sensitivity to changes in the training dataset. Therefore, high variance can lead to overfitting, meaning the model performs exceptionally well on training data but generalizes poorly on unseen data. Meanwhile, the variance due to the irreducible error represents the inherent noise in the data that cannot be reduced regardless of the model’s complexity or optimization. This type of error arises from intrinsic limitations in data collection processes, measurement inaccuracies, or underlying stochastic variability in the observed phenomena. As a result, even an optimally trained model will have a lower bound on its predictive accuracy due to these unavoidable sources of uncertainty. Consequently, achieving a trade-off (an optimal balance between bias and variance) is critical for building robust models. Overemphasis on one often exacerbates the other.

Furthermore, the literature identified problems with the datasets used to build reliable, robust models, for instance, in models using imbalanced datasets in thyroid cancer classification (e.g., minority classes like rare tumor types). These models presented a hindered ability to capture patterns and classify the minority class accurately. Imbalance often skewed predictions toward the majority class and reduced the model’s sensitivity to less common outcomes. Some papers addressed class imbalance-related problems in the recurrence data [11,12,18] and resolved them by applying different sampling techniques, most often oversampling, by increasing minority class examples (e.g., SMOTE). However, it is acknowledged that it may introduce noisy and uninformative samples [11,18] and challenges due to poor generalization, overfitting, and limited handling of complex and diverse datasets. Moreover, the inconsistency in dataset selection is another significant challenge in evaluating and comparing the studies that target AI in oncologic endocrinology. Many researchers relied on entirely different datasets, making it difficult to establish a standardized benchmark for assessing relative performance. Even in studies that theoretically used the same database, such as SEER, the exact employed subset often differed, further complicating direct comparisons. This issue is exacerbated because a substantial part of these datasets is not publicly available, impeding reproducibility and preventing comprehensive cross-study comparisons. Consequently, the field lacks a unified evaluation framework, which hinders meaningful progress and limits the generalizability of findings.

Additionally, most of the reviewed works were single-center observational studies, introducing a potential selection bias that may limit the generalizability of findings. Therefore, concerted efforts are needed to enhance data availability and quality, as the lack of large, high-quality datasets remains a significant barrier to developing robust AI models for endocrine cancer diagnosis. One key strategy for improving ML applications in this field is the enhanced sharing of existing data by establishing open-access databases for various disease types or fostering collaborative efforts across institutions to aggregate diverse datasets. Creating comprehensive and well-annotated datasets is crucial for training and validating AI algorithms, improving their reliability and clinical utility. Currently, several thyroid cancer databases contribute to ML-driven diagnosis, prognosis, monitoring, and treatment [60]. These include the TNM8 Dataset (Guidelines) [61], SEER Database [44], Digital Database Thyroid Image (DDTI Dataset) [62], Gene Expression Omnibus (GEO Repository) [63], and Cancer Genome Atlas (TCGA Project) [64]. While these databases are valuable, more comprehensive and complex datasets are needed to further advance AI applications in thyroid cancer research.

Another aspect to be discussed is that most studies have employed similar methodological approaches. However, only a few experimented with a broader range of models or more sophisticated preprocessing techniques. While some studies explored alternative architectures or fine-tuned their preprocessing pipelines, the overall methodological landscape remained largely homogeneous. Notably, attempts to leverage more powerful models such as Transformers [65] were absent, except for Sievert et al. [20], who explored the use of LLMs. This observation suggests a reluctance or a lack of resources to engage with state-of-the-art deep learning architectures. Such a situation needs attention and solutions that could enhance performance and robustness.

Furthermore, ensuring the seamless integration of these technologies into routine medical workflows requires fostering physician acceptance and trust. It is generally acknowledged that the black-box nature of many ML models is a major limitation that diminishes trust among clinicians and patients. Therefore, developing transparent and explainable AI models is imperative, as it will enable clinicians to interpret and validate predictions with confidence. Moreover, the absence of interpretability in ML-driven decision-making could lead to non-compliance with regulatory standards, as many healthcare regulations mandate transparency, accountability, and explainability in medical decision-support systems. While it is true that deep learning models provide less interpretability compared to classical ML approaches, avoiding their use can significantly limit both the types of data that can be effectively utilized and the level of performance that could be achieved. Ensuring interpretability is, therefore, not only a technical necessity but also a regulatory and ethical requirement for the successful deployment of AI in oncologic endocrinology.

Meanwhile, the variability in clinical evaluation criteria and heterogeneous sample sizes observed across the investigated studies are worth noting from an ethical point of view. As the number of participants and data-sharing initiatives increases, it is essential to implement robust cybersecurity measures to mitigate potential cyber risks and safeguard sensitive patient information from unauthorized access or cyberattacks. Data security and patient privacy should be a priority in developing AI-driven healthcare solutions.

Another essential aspect to be discussed in connection with patient data is that across all reviewed studies, there is an almost complete lack of data regarding the history of thyroid-related medication, other treatments, and comorbidities. Among them, only Kim et al. [22] provided comprehensive data on thyroid cancer patients who underwent bilateral total thyroidectomy, received radioiodine therapy, and were followed for over five years. This study specifically collected data on the frequency of radioiodine ablation therapy and radiation doses administered, highlighting a crucial gap in existing research. This limitation is further emphasized in a comprehensive bibliometric analysis of thyroid cancer (TC) research spanning three decades (1990–2020) [66]. The study aimed to analyze 30 years of scientific literature on TC using latent Dirichlet allocation, an ML-based topic modeling technique. Data were sourced from PubMed using the MeSH term thyroid neoplasms, filtered for English-language publications, yielding 34,692 articles analyzed, averaging 1119 publications per year. The top five contributing countries in TC research were China, the USA, Italy, South Korea, and Japan. The analysis categorized the literature into four major research clusters: (a) treatment management; (b) basic research; (c) diagnosis research; and (d) epidemiology and cancer risk. Notably, the study reported that, in recent years, research focused on active surveillance in TC, an observation reflected in the top 10 most cited publications.

However, an important gap was the weak linkage between basic research and treatment management, suggesting that greater efforts are needed to translate fundamental scientific discoveries into clinical therapies and treatment strategies [66,67] to improve TC patients’ Quality of Life (QoL). From a clinical and economic epidemiology point of view, another critical concern is TC management. Regarding Quality of Life (QoL) metrics, Liu et al. [68] developed an RF-based model to predict QoL decline in thyroid cancer patients three months post-thyroidectomy. The study identified the seven most significant factors influencing post-surgical QoL: clinical stage, marital status, histological type, age, nerve injury symptoms, economic income, and surgery type. Although this study provides valuable insights, further research is required to enhance the understanding of QoL determinants and to refine predictive models for post-treatment patient well-being.

Towards this end, Li et al. [69] investigated the QoL of patients who underwent thyroidectomy for lower-risk PTC, as opposed to those who chose active surveillance to monitor the disease actively. The authors identified that 24.2% of the patients in the group who underwent the surgery showed heightened regret as opposed to only 3.4% in the surveillance group. They found that the large majority of regrets (75%) stemmed from postoperative QoL issues, such as scarring (i.e., cosmetic concerns affecting confidence and social interactions), psychological distress (e.g., anxiety, depression, or dissatisfaction with the decision), and neuromuscular symptoms (e.g., fatigue, muscle weakness). Since machine intelligence continues to gain attention in oncologic endocrinology due to its potential to enhance robust, non-invasive diagnostics, the technology must be effectively adopted in clinical practice through standardized reporting guidelines and rigorous evaluation criteria. The wide adoption of ML towards the development of personalized therapy could enhance the QoL of patients, especially for those with low-risk cases where surgery may not have been necessary.

A summary containing the highlights of this section is presented in Box 2.

Box 2Challenges and limitations highlights.

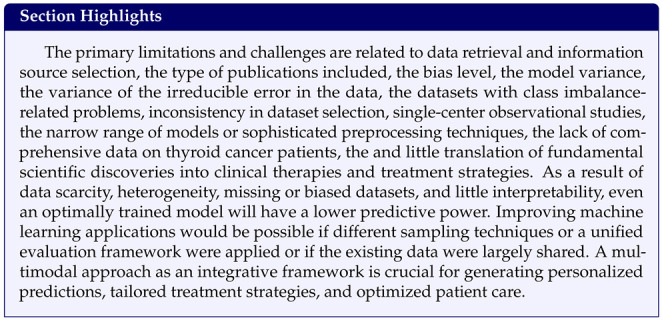



### 4.3. Perspectives and Future Directions

As argued in this scoping review, ML greatly impacts thyroid cancer research, particularly in areas such as cancer detection, malignancy prediction, prognostic system development, recurrence, and survival prediction. Despite its promising applications, several challenges persist in leveraging ML for thyroid cancer, including data scarcity, heterogeneity, missing or biased datasets, and issues related to model interpretability. More longitudinal studies are necessary to advance the field, emphasizing expanded datasets and long-term follow-ups to enhance the validation and refinement of predictive models. Additionally, a multimodal approach that integrates imaging, clinical, genomic data, lifestyle, and sociodemographic variables is essential for generating personalized predictions, tailored treatment strategies, and optimized patient care. Such an integrative framework is crucial for achieving a comprehensive, data-driven understanding of thyroid cancer and improving clinical decision-making in the long term.

The global digital medicine market in the United States is projected to exceed 500 billion dollars by 2025 [70]. Digital medicine transforms healthcare by integrating technology and medicine to enhance healthcare delivery, patient outcomes, and overall system efficiency. This field encompasses diverse intelligent and interconnected methodologies and tools, including wearable devices, mobile applications, and sensors for real-time health monitoring, as well as the remote delivery of healthcare services via video consultations, phone calls, and digital platforms. Integrating digital tools, like picture archiving, and communication systems, and electronic health records, has facilitated data storage, multimodal access, and seamless network transmission, enabling improved clinical workflows. Simultaneously, advancements in biosensors and AI/ML technologies have propelled digital medicine forward, enhancing diagnosis, treatment, monitoring, and prevention strategies. However, alongside these innovations, ensuring patient privacy, ethical considerations, and regulatory compliance remains a critical priority for developing robust and interpretable AI models suited for clinical implementation. Overcoming these challenges will require interdisciplinary collaboration among clinicians, data scientists, and policymakers, ensuring the successful integration of ML tools into routine clinical practice. This paper associated the current knowledge in thyroid cancer care with the innovative approaches incorporating ML into its management. Ultimately, these advancements help improve diagnosis and treatment in endocrinology, leading to more precise, personalized, and data-driven care for patients (Figure 3).

In clinical practice, ML tools have the potential to assist medical practitioners in making faster, more accurate diagnostic decisions, to help avoid unnecessary biopsies by better identifying which patients are indeed at risk, and to allow for more personalized treatment plans that are tailored to each patient’s unique characteristics. For example, predictive models integrated into electronic health records could alert clinicians to high-risk patients, enabling earlier intervention. In postoperative care, recurrence prediction models may guide the intensity of follow-up, making better use of medical resources while also giving patients greater peace of mind. These applications not only improve the efficiency of clinical workflows but also promote more collaborative and informed decision-making between clinicians and patients. However, successful implementation will require thorough validation in real-world settings, clinician training, and careful consideration of economic, legal, and ethical responsibilities.

## 5. Conclusions

The use of artificial intelligence to predict clinical outcomes in thyroid cancer has grown exponentially over the past decade. We conducted a literature analysis of six online databases, following the PRISMA-ScR guidelines, for English-language publications between 2014 and 2024 that studied ML applications in thyroid cancer. Ultimately, 21 of the most relevant full-text articles were included and presented in this scoping review.

Machine learning techniques have immense potential in thyroid cancer research, offering novel solutions for diagnosis, metastasis prediction, prognosis, and treatment personalization. Beyond these applications, ML holds promise in drug discovery, personalized treatment strategies, and long-term patient monitoring, areas that could significantly transform thyroid cancer care. While challenges remain, advancements in computational methods, dataset accessibility, and algorithm transparency will further enhance ML’s role in precision oncology. Future studies should emphasize interdisciplinary collaboration between clinicians and data scientists to maximize ML’s impact on thyroid cancer management and expand its applications in emerging areas of personalized medicine.

This work summarizes existing knowledge in thyroid cancer care and discusses the challenges and opportunities of integrating ML tools into routine endocrinology clinical workflows. By addressing these aspects, we seek to contribute to the growing dialogue on integrating ML into clinical practice to improve thyroid cancer management, ultimately leading to precision and personalized medicine.

## Figures and Tables

**Figure 1 cancers-17-01308-f001:**
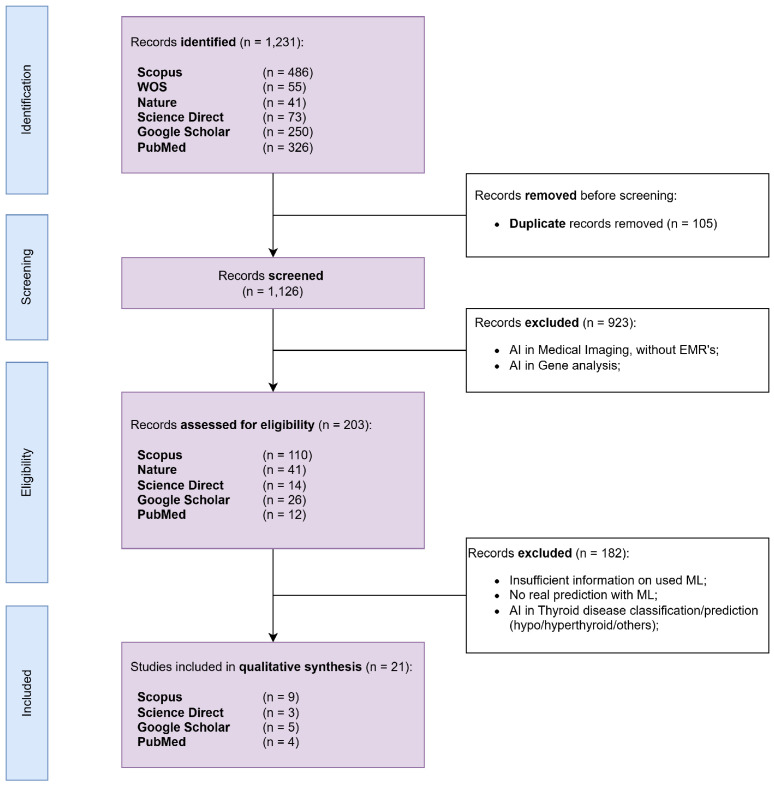
PRISMA flow diagram.

**Figure 2 cancers-17-01308-f002:**
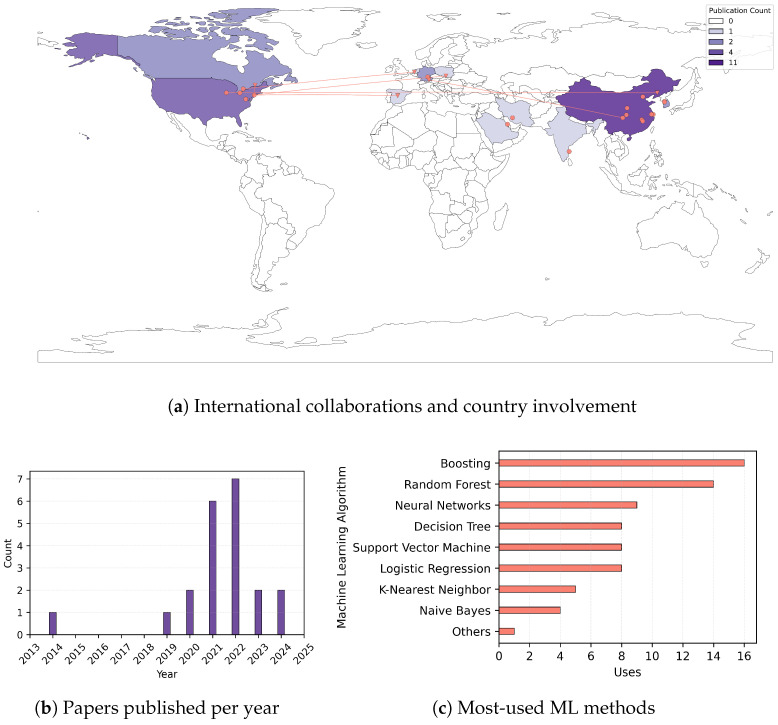
Selected studies’ statistics.

**Figure 3 cancers-17-01308-f003:**
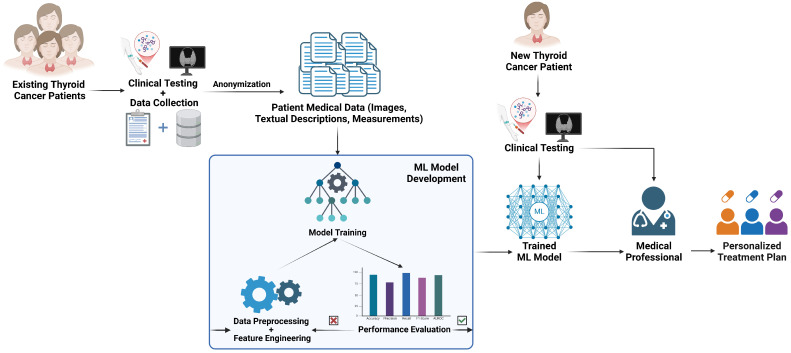
Framework for personalized patient care using machine learning.

**Table 1 cancers-17-01308-t001:** Eligibility criteria for text screening.

Inclusion	Exclusion
Research articles and conference papers written in English that are fully available online	Review articles, study protocols, book chapters, notes, brief reports, letters, editorials, or case studies written in any language
Published in Scopus, Web of Science, Nature, Science Direct, Google Scholar, or PubMed between 2014 and 2024 with more than 10 citations between 2014–2022 and at least one between 2023 and 2024	Articles focusing on thyroid biomarkers, gene expressions, hypothyroidism, or hyperthyroidism
Participants: Adults and children with current thyroid cancer diagnosis or just suspicions	Articles using ultrasound images or radiomics generated from images
Concept: Patients undergoing blood analyses, testing, and other imaging tests features in hospitals or input data from repositories for detecting disease progression	
Objective: Machine learning applied to medical data to make predictions and prognostics for thyroid cancer	
Context: Feature thyroid cancer management integrating ML tools into routine clinical workflows	

**Table 2 cancers-17-01308-t002:** Included studies.

Ref	Type	Input Data	Objective	BPC
[10]		blood test results, TNM stage, ultrasound features, surgical methods	predicting thyroid nodule malignancy	RF
[11]	PTC	TNM stage, histological type, surgical methods	predict PTC recurrence	DT
[12]	PTC, FTC, MTC, ATC	TNM stage, histological type	predict lung metastasis in TC	RF
[13]	PTC, FTC	TNM stage, histological type, regional nodes examined, survived months	survival rate (for over 10 years since diagnosis)	MLP
[14]		TNM stage, histological type, type of treatment	survival prediction in TC patients	MLP
[15]	PTC, FTC, MTC, ATC	TNM stage, histological type	predict bone metastasis in people with TC	RF
[16]	PTC	TNM stage, histological type, ultrasound features, surgical methods	predict CLNM	XGBoost
[17]	PTC	blood test results, ultrasound features, surgical methods	predict CLNM	GBDT
[18]		ultrasound features	TC prediction models, malignancy prediction	DT
[19]		blood test results	detect TC at very early stages	RF
[20]		ultrasound features	classify sonographic patterns in accordance with TI-RADS	LLM
[21]	PTC	blood test results, TNM stage, and ultrasound features	predict LLNM in PTC patients	RF
[22]	Well-Differentiated TC	pathological and genetic information	predicting disease recurrence	ILP
[23]	PTC (≤1 cm)	family history of cancer, blood test results, pathological features, ultrasound features	predict the risk of CLNM	RF
[24]	PTC	TNM stage, histological type, ultrasound features	predict malignant nodules in PTC	RF
[25]		TNM stage, pathology and cytology data, ultrasound features	predict malignancy in indeterminate thyroid nodules	RF
[26]	Well-Differentiated TC	TNM stage, histological type, follow up vital status	prognostic systems for well-differentiated TC	PAM
[27]	FTC	TNM Stage, histological type, surgical methods	predict the prognosis of FTC	XGBoost
[28]	PTC	TNM stage, histological type, ultrasound features, genetic information, surgical methods	predict CLNM in PTC	CNN
[29]	PTC	TNM stage, histological type	predict LLNM of PTC without central lymph node metastasis	SVM
[30]	PTC	blood test results, TNM Stage, ultrasound features	predict DLNM in PTC patients	RF

Ref = reference; Type = type of Cancer; BPC = best performing classifier. Input data summarizes other data except demographics. All other abbreviations may be found in the Abbreviations Table.

## Data Availability

The material used in this study is available from the source databases (accessed in November 2024). The study protocol was registered on INPLASY (INPLASY202520118).

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
