# Peer review of "Machine Learning for Thyroid Cancer Detection, Presence of Metastasis, and Recurrence Predictions—A Scoping Review"

_cancers, 2025, doi:10.3390/cancers17081308_

Round 1

Reviewer 1 Report

Comments and Suggestions for Authors

This is a systematic review (called scoping review) of machine learning in thyroid cancer that covers articles published between 2014-2024.  The selection of articles followed the PRISMA process starting with >1200 articles and ending by 21 articles that cover three domains; early detection of thyroid cancer, prediction of distant metastases and prediction of recurrence of thyroid cancer.  The methods used in each article was listed and the articles summarized under appropriate domains.  The discussion was relevant and future perspectives and limitations were discussed. Overall, the review is of good quality and touches upon a very expanding and active field of AI and machine leaning that is progressively entering clinical arena.

My comments relate to:

  1. The manuscript has significant technical terminology and methodologies that are difficult to understand by the casual reader.  One of them is the methods used by different articles (e.g. Boosting, Random Forest...etc).  I would suggest a brief and simplified description of these methods in a supplementary text or table so the reader can follow the text in the main manuscript
  2. Because the manuscript is lengthy, I suggest short summary at the end of each domain, 3.2.1, 3.2.2 and 3.2.3)
  3. Line 17-20: there is redundancy in this part, please try to improve it
  4. Line 25: scoring or scoping review?
  5. Line 44: Hurthle cell cancer is a term that is no more used and replaced by Oncocytic thyroid cancer, please change.  Furthermore, Oncocytic thyroid cancer is not considered anymore a variant of follicular thyroid cancer.  It is a distinct entity as per the latest WHO classification
  6. Line 64: actionable insights is not a clear term, please rephrase
  7. It is advisable that the manuscript is shortened as much as possible to facilitate reading and understanding by regular medical personnel

Author Response

Comment 0: This is a systematic review (called scoping review) of machine learning in thyroid cancer that covers articles published between 2014-2024.  The selection of articles followed the PRISMA process starting with >1200 articles and ending by 21 articles that cover three domains; early detection of thyroid cancer, prediction of distant metastases and prediction of recurrence of thyroid cancer.  The methods used in each article was listed and the articles summarized under appropriate domains.  The discussion was relevant and future perspectives and limitations were discussed. Overall, the review is of good quality and touches upon a very expanding and active field of AI and machine leaning that is progressively entering clinical arena.

Response 0: Thank you kindly for your positive feedback and thorough review.

My comments relate to:

Comment 1: The manuscript has significant technical terminology and methodologies that are difficult to understand by the casual reader.  One of them is the methods used by different articles (e.g. Boosting, Random Forest...etc).  I would suggest a brief and simplified description of these methods in a supplementary text or table so the reader can follow the text in the main manuscript.

Response 1: Thank you for your suggestion! You are right; it may be difficult for a reader to follow the manuscript without prior knowledge of these methods. As such, we have introduced a section in the appendix that provides a concise explanation of the ML methods used in the identified articles.

Comment 2: Because the manuscript is lengthy, I suggest short summary at the end of each domain, 3.2.1, 3.2.2, and 3.2.3)

Response 2: Thank you for your suggestion. It is true that given the fact that each domain is fairly lengthy, a summary at the end of each would be useful for the reader. As such, we have provided a concise overview at the end of sections 3.2.1, 3.2.2, and 3.2.3, to ensure that readers can easily recall the key points discussed in each section. Additionally, we provided two boxes containing section highlights, one for section 3.2 and one for section 4.2 (the one regarding challenges and limitations)

Comment 3: Line 17-20: there is redundancy in this part, please try to improve it

Response 3: Thank you for your keen observation! We have modified the abstract to eliminate the redundancy while keeping all necessary information.

Comment 4: Line 25: scoring or scoping review?

Response 4: Thank you for your keen eye! We have corrected the mistake!

Comment 5: Line 44: Hurthle cell cancer is a term that is no more used and replaced by Oncocytic thyroid cancer, please change.  Furthermore, Oncocytic thyroid cancer is not considered anymore a variant of follicular thyroid cancer.  It is a distinct entity as per the latest WHO classification

Response 5: Thank you for your observation! Indeed, our classification was outdated, as such we have updated it according to the 5th edition of the World Health Organization classification of Endocrine and Neuroendocrine Tumors.

Comment 6:  Line 64: actionable insights is not a clear term, please rephrase

Response 6: Thank you for your observation! Indeed, actionable insights were a pretty vague term. We have replaced it with a more suggestive expression: “meaningful patterns and valuable information”

Comment 7:  It is advisable that the manuscript is shortened as much as possible to facilitate reading and understanding by regular medical personnel

Response 7: Thank you for your suggestion! We acknowledge that the manuscript's length may pose a challenge for some readers. However, our goal is to provide a thorough analysis, ensuring that key insights are not lost. While a more concise version may enhance readability, it could compromise the depth and clarity of the discussion. We have carefully structured the manuscript to balance readability with the need for a complete and meaningful presentation of the findings, to the best of our ability. Additionally, we have added two boxes that contain highlights of the most important information for some our our most important sections, namely section 3.2 where we discuss the relevant articles, and section 4.2 where we discuss the challenges and limitations. We hope that these two boxes will aid in giving an overview of the important information for readers who are not able to read the entire sections.

Reviewer 2 Report

Comments and Suggestions for Authors

The review “Machine Learning for Thyroid Cancer Detection, Presence of Metastasis, and Recurrence Predictions - A Scoping Review” summarized the current applications of ML in thyroid cancer, including methodologies and their performance, and investigated how ML is used to develop simulation and prediction strategies based on clinical data and electronic medical records. With the increasing number of thyroid cancers, this area of research is of great clinical importance. Here are a few comments for the authors’ consideration:

  1. In the INTRODUCTION section, it is recommended to elaborate on thyroid cancer before further elaborating on deep learning to make the manuscript's content better organized.
  2. Lines 72-75, “We highlight the ML methods used, discuss their performance through metrics like accuracy, specificity, sensitivity, or F1-score, and compare them with traditional statistical and diagnostic methods.”Indicators for specific assessment models are preferably presented in the methodology section.
  3. Lines 541-543, “Furthermore, ML impacts medical education through AI-driven decision support systems and clinical applications for risk stratification and treatment planning.”How has medical education been affected?
  4. lines 706-709, “However, an important gap was the weak linkage between basic research and treatment management, suggesting that greater efforts are needed to translate fundamental scientific discoveries into clinical therapies and treatment strategies [61,62] to improve TC patients' quality of life (QoL). From clinical and economics epidemiology points of view, another critical concern is TC management.”This latest article (PMID: 40057484) also studied QoL in post-thyroidectomy patients, please refer to and cite it.
Comments on the Quality of English Language

The English could be improved to more clearly express the research.

Author Response

Comment 0: The review “Machine Learning for Thyroid Cancer Detection, Presence of Metastasis, and Recurrence Predictions - A Scoping Review” summarized the current applications of ML in thyroid cancer, including methodologies and their performance, and investigated how ML is used to develop simulation and prediction strategies based on clinical data and electronic medical records. With the increasing number of thyroid cancers, this area of research is of great clinical importance. Here are a few comments for the authors’ consideration:

Response 0: Thank you for your thorough feedback!

Comment 1: In the INTRODUCTION section, it is recommended to elaborate on thyroid cancer before further elaborating on deep learning to make the manuscript's content better organized.

Response 1: Thank you for your suggestion! We agree that the introduction could have been better structured; as such, we have slightly reorganized it, while also adding additional information on thyroid cancer.

Comment 2: Lines 72-75, “We highlight the ML methods used, discuss their performance through metrics like accuracy, specificity, sensitivity, or F1-score, and compare them with traditional statistical and diagnostic methods.”Indicators for specific assessment models are preferably presented in the methodology section.

Response 2: Thank you for your observation! Indeed, while we reported different metrics (accuracy, AUROC, and F1), we never introduced them and explained what each effectively indicates. To correct this omission, we have added a paragraph in the “Materials and Methods” section that introduces the metrics that were used, along with a concise explanation for each of them.

Comment 3: Lines 541-543, “Furthermore, ML impacts medical education through AI-driven decision support systems and clinical applications for risk stratification and treatment planning.”How has medical education been affected?

Response 3: Thank you for your question! The lines you highlighted contained two typos. While the sentence was phrased as a conclusion, it was supposed to be hypothetical. Additionally, we originally meant to refer to medical practice, not medical education. To clarify this, we have revised the line as follows: “... ML could improve medical practice through …”.  You are correct that while the studies we presented suggest that integrating ML into medical practice may yield positive outcomes for patients, there is still limited documented evidence on how this integration directly impacts medical education. Furthermore, there is insufficient research to determine whether overreliance on these methods could create challenges for medical personnel training in the long term.

Comment 4: lines 706-709, “However, an important gap was the weak linkage between basic research and treatment management, suggesting that greater efforts are needed to translate fundamental scientific discoveries into clinical therapies and treatment strategies [61,62] to improve TC patients' quality of life (QoL). From clinical and economics epidemiology points of view, another critical concern is TC management.”This latest article (PMID: 40057484) also studied QoL in post-thyroidectomy patients, please refer to and cite it.

Response 4: Thank you very much for your suggestion. We’ve reviewed the article you provided and incorporated relevant information from it to provide a more comprehensive understanding of how thyroidectomy affects the QoL of patients, and integrating data-driven approaches towards personalized treatment could aid.

Reviewer 3 Report

Comments and Suggestions for Authors

This issue   is current, congratulations.
First suggestion:
do not use abbreviations
Second recommendation:"Thyroid cancers can be differentiated, poorly differentiated, and undifferentiated. Differentiated cancers include several forms: a) papillary, which is the most frequent andleast aggressive; b) follicular, which is the second most frequent; c) Hürttle cell, a rare type of follicular form; and d) medullary, which is associated with a genetic mutation. Undifferentiated thyroid cancer is typically anaplastic, the most aggressive form of thyroidcancer, rare and difficult to treat [1,2]. Papillary (PTC) and follicular (FTC) carcinomas (with their variants) generally have a good prognosis, with 10-year survival rates exceeding 90% [1]. Medullary carcinoma is more aggressive than the first two and treated more precisely than the other types." You used a classification that has already been UPDATED IN 2023. Please review it 
"In this work, we surveyed existing literature on existing ML applications in thyroid cancer based on patient data. We also evaluated the role of ML models in areas that couldlead to personalized medicine, such as interpreting thyroid cancer diagnoses, predicting malignancy, improving thyroid cancer early detection, or predicting recurrence. We highlight the ML methods used, discuss their performance through metrics like accuracy, Version March 2, 2025 submitted to Cancers 3 of 24
specificity, sensitivity, or F1-score, and compare them with traditional statistical and diagnostic methods. Additionally, we discuss the advantages, limitations, challenges, ethicalconsiderations, and future directions toward personalized medicine" -what is the central question? It was not clear and this impacts your conclusion ( there is no sharp answer)

Your research methods and selection are very clear and, in my opinion, adequate, but the limitation of the works in English brings a limitation to the work that should be pointed out in the discussion (I did not see this comment in your discussion).
Therefore, table 1 is in results and it should be brought to methods
Your results need to be grouped so that you can say with this the current situation. Work on this, it will improve your paper 

Finally, try to ask a clear question and answer it there. Your work was more like an overview of the current situation of artificial intelligence than its use itself. And it is what we reader expect 

Author Response

Comment 0: This issue   is current, congratulations.

Response 0: Thank you kindly for your positive feedback and detailed review.

Comment 1: First suggestion:
do not use abbreviations

Response 1: Thank you for your suggestion. While we think removing all abbreviations would elongate the manuscript more than necessary, we agree that some abbreviations we used were unnecessary, as such we have removed the use of abbreviations in the cases where that abbreviation appeared 3 times or less.
Comment 2: Second recommendation:"Thyroid cancers can be differentiated, poorly differentiated, and undifferentiated. Differentiated cancers include several forms: a) papillary, which is the most frequent andleast aggressive; b) follicular, which is the second most frequent; c) Hürttle cell, a rare type of follicular form; and d) medullary, which is associated with a genetic mutation. Undifferentiated thyroid cancer is typically anaplastic, the most aggressive form of thyroidcancer, rare and difficult to treat [1,2]. Papillary (PTC) and follicular (FTC) carcinomas (with their variants) generally have a good prognosis, with 10-year survival rates exceeding 90% [1]. Medullary carcinoma is more aggressive than the first two and treated more precisely than the other types." You used a classification that has already been UPDATED IN 2023. Please review it

Response 2: Thank you for the keen observation! We have updated our classification according to the 5th edition of the World Health Organization classification of Endocrine and Neuroendocrine Tumors.

Comment 3: "In this work, we surveyed existing literature on existing ML applications in thyroid cancer based on patient data. We also evaluated the role of ML models in areas that couldlead to personalized medicine, such as interpreting thyroid cancer diagnoses, predicting malignancy, improving thyroid cancer early detection, or predicting recurrence. We highlight the ML methods used, discuss their performance through metrics like accuracy, Version March 2, 2025 submitted to Cancers 3 of 24
specificity, sensitivity, or F1-score, and compare them with traditional statistical and diagnostic methods. Additionally, we discuss the advantages, limitations, challenges, ethicalconsiderations, and future directions toward personalized medicine" -what is the central question? It was not clear and this impacts your conclusion ( there is no sharp answer)

Response 3: Thank you for your advice! While we had a main direction while we were creating this scoping review, indeed, we did not explicitly write what the central question guiding our work was. To address this, we have modified our introduction to offer a clearer view of our motivation.

Comment 4: Your research methods and selection are very clear and, in my opinion, adequate, but the limitation of the works in English brings a limitation to the work that should be pointed out in the discussion (I did not see this comment in your discussion).
Therefore, table 1 is in results and it should be brought to methods
Your results need to be grouped so that you can say with this the current situation. Work on this, it will improve your paper 

Response 4: Thank you for your positive feedback! Regarding the improvements you suggested, in the discussion section, the English language limitation is discussed in the second paragraph of section 4.2 we discuss both the limitations of our study, as well as those identified in the reviewed articles. To make the distinction clearer, we slightly altered the section. Regarding the table, you are right, we referenced the table in section 2, and we wanted it to be displayed there, but because of the figure before it appeared in the PDF right after the start of section 3. With the new modifications done in this review round, it now appears before section 3.

Comment 5: Finally, try to ask a clear question and answer it there. Your work was more like an overview of the current situation of artificial intelligence than its use itself. And it is what we reader expect 

Response 5: Thank you for your suggestion, we have added a research question that served as the main direction for this review in the introduction section.

Reviewer 4 Report

Comments and Suggestions for Authors

Dear Authors, 
this is a really interesting and well developed paper that focuses on an open field of research. The quality of the manuscript is overall good, I have only some minor suggestions and requests:
- lines 124-126 "Articles were excluded if they solely incorporated patient imaging in their data collection or applied ML to biomarkers or thyroid cancer-related genes": the meaning of this sentence is hard to understand and since it is a fundamental part of the manuscript it needs to be clarified;
- an evaluation of the papers included in the review, for example with the QUADAS tool, will strengthen the overall quality of the paper;
- a wider discussion on the impact of ML on the clinical practice and on the management of the patients will strengthen the overall quality of the paper;
- reproducibility of models and findings is in general an issue that is present in all of the studies that focuses on ML. This should be described and discussed;
- a brief discussion on the role of imaging-based ML in thyroid cancer (for example: 10.1007/s11154-023-09822-4) will strengthen the overall quality of the paper.

Author Response

Comment 0: Dear Authors, 

this is a really interesting and well developed paper that focuses on an open field of research. The quality of the manuscript is overall good, I have only some minor suggestions and requests:

Response 0: Thank you kindly for your positive feedback and thorough review!

Comment 1: - lines 124-126 "Articles were excluded if they solely incorporated patient imaging in their data collection or applied ML to biomarkers or thyroid cancer-related genes": the meaning of this sentence is hard to understand and since it is a fundamental part of the manuscript it needs to be clarified;

Response 1: Thank you for your suggestion. We have modified the sentence to enhance its clarity. Now the sentence is the following: “Articles were excluded for any of the following: (1) it only used patient imaging data; (2) it focused on applying machine learning to biomarkers or thyroid cancer-related genes.”

Comment 2: - an evaluation of the papers included in the review, for example with the QUADAS tool, will strengthen the overall quality of the paper;

Response 2: Thank you for your valuable suggestion! We agree that a formal quality assessment framework such as QUADAS could enhance the methodological rigor of the paper. However, since our scoping review covers a broad range of ML applications beyond diagnostic accuracy and statistical tests, we believe the QUADAS tool may not be suitable for many of the included studies. To our knowledge, QUADAS targets systematic literature reviews, not scoping reviews. Additionally, adding this type of evaluation would require significantly expanding the scope and goals of our current study. Nevertheless, we acknowledge the importance of quality assessment and have taken care to highlight the included studies' limitations, strengths, and methodological considerations throughout the review.

Comment 3: - a wider discussion on the impact of ML on the clinical practice and on the management of the patients will strengthen the overall quality of the paper;

Response 3: Thank you for your suggestion! While we believe that section 4.3 included a relevant discussion on this topic, we agree that a broader perspective could enhance the paper. As such we have added an additional paragraph that addresses this point in greater detail.

Comment 4: - reproducibility of models and findings is in general an issue that is present in all of the studies that focuses on ML. This should be described and discussed;

Response 4: Thank you for your observation! This is indeed a major problem in the majority of the papers we identified, as such, this was discussed in section 4.2, more exactly, the paragraph regarding the datasets used. It is the part that starts the following sentence: “Moreover, the inconsistency in dataset selection is another significant challenge in evaluating and comparing the studies that target AI in oncologic endocrinology.”

Comment 5: - a brief discussion on the role of imaging-based ML in thyroid cancer (for example: 10.1007/s11154-023-09822-4) will strengthen the overall quality of the paper.

Response 5: Thank you for your suggestion! We conceptually agree, but this is a specific exclusion criterion for our paper - as such, we feel that it would be confusing for the readers. If you prefer, we can add a paragraph in the conclusions with future research directions, covering also image processing, but that is a different study.

Round 2

Reviewer 1 Report

Comments and Suggestions for Authors

The authors have improved the manuscript, but I still have two minor comments.  

  1. There is no need to list all types of thyroid cancer according to the WHO classification as this consumes space and is not well connected with the subject of the manuscript (Machine learning).  Enough to say "thyroid follicular cell-derived thyroid cancer is classified into 1. differentiated comprised of papillary, follicular and oncocytic types and their subtypes and 2. undifferentiated (commonly called anaplastic) thyroid cancer
  2. In the Section Highlights, page 17, third paragraph: "Malignancy models relied on histological features like nodule size, shape, location, multifocality, and calcification" These seem to me radiological features and not histological features or a combination of both.  Please check
  3. I still think the manuscript is too lengthy to be read without pain! 

Author Response

Comment 0: The authors have improved the manuscript, but I still have two minor comments.  

Response 0: Thank you for your positive feedback!

Comment 1: There is no need to list all types of thyroid cancer according to the WHO classification as this consumes space and is not well connected with the subject of the manuscript (Machine learning).  Enough to say "thyroid follicular cell-derived thyroid cancer is classified into 1. differentiated comprised of papillary, follicular and oncocytic types and their subtypes and 2. undifferentiated (commonly called anaplastic) thyroid cancer

Response 1: Thank you for your suggestion! We have modified that section to mention only the relevant types of thyroid cancer in the context of the studies presented in our article, while still maintaining the classification according to the WHO guidelines.

Comment 2: In the Section Highlights, page 17, third paragraph: "Malignancy models relied on histological features like nodule size, shape, location, multifocality, and calcification" These seem to me radiological features and not histological features or a combination of both.  Please check

Response 2: Thank you for your observation! You are correct that the features can be both histological and radiological, depending on the modality. Since the paragraph begins with a reference to "biochemical, ultrasound, and histological data," we have revised the sentence to clarify the nature of the data used. Specifically, we have removed the word “histological” from the sentence, allowing each referenced article to specify the data type used for the features they assessed.

Comment 3: I still think the manuscript is too lengthy to be read without pain! 

Response 3: Thank you for your feedback! We understand that the length of the manuscript may be overwhelming for some readers; however, this is a complex topic, and we wanted to ensure that all the information we deem necessary was included. We’ve made explicit efforts to be concise while still delivering the key points, and we believe that reducing the size to an easily digestible length would lead to the loss of important details. We also considered moving part of the information to an Appendix, but we did not enjoy the new presentation, which seemed to lack depth and key details on each study.

Reviewer 3 Report

Comments and Suggestions for Authors

The work is now adequate, in a short time they were able to review it and give it a more up-to-date look (classifications), structure a question, have conclusions that respond to the objective and point out limitations.
I think the work can be published.

Author Response

Comment 1: The work is now adequate, in a short time they were able to review it and give it a more up-to-date look (classifications), structure a question, have conclusions that respond to the objective and point out limitations.

I think the work can be published.

Response 1: Thank you kindly for your positive feedback!